# OpenReview forum: "Enhanced SAM-Med3D: A Robust Solution for 3D Medical Image Segmentation with Advanced Post-processing"
_thecvf.com/CVPR/2025/Workshop/MedSegFM — CVPR 2025 Workshop MedSegFM Submission_

### Official Review · Reviewer_v6P5 · 2025-09-13
**The paper should be rejected, it proposes a novel 5 step post-processing pipeline as well as a custom SAM-Med3D training pipeline, but fails to demonstrate how either of these contributions is an improvement over the existing baselines.**

**Rating:** 3
**Confidence:** 4

**Review:**

The main claim of the paper is that their novel 5 step post-processing technique combined with their custom training pipeline represents an enhancement over the SAM-Med3D model.
The authors modified the original SAM-Med3D architecture by halving the size of the ViT embedding dimensions, they also used different learnings rates for different parts of the architecture as well as a multi-step learning rate scheduler and gradient accumulation which they motivated by the limited data setting.
In their post-processing pipeline they applied region filtering, hole filling, morphological closing, gaussian smoothing and resolved overlaps.
The model presented in the paper significantly underperforms all baselines it was compared against, both with and without their custom post-processing pipeline. Neither version was able to beat the original SAM-Med3D or any of the other baselines in any metric on any modality.
One potential reason for this is that the authors were limited to 10% of the training data due to participating in the corest track. A comparison to other methods pre-trained only on the same data, as well as an ablation of the original SAM-Med3D model using the 5 step post-processing pipeline would have been interesting.
The authors compared their model with preprocessing to a version without, the custom preprocessing they present performed worse in all modalities except PET, where there was a slight gain in performance.

The paper should be rejected, it proposes a to my knowledge novel 5 step post-processing pipeline as well as a custom SAM-Med3D training pipeline, but fails to demonstrate how either of these contributions is an improvement over the existing baselines. The only case where their post-procesisng pipeline was helpful was in the PET modality and even there the original SAM-Med3D was significantly outperforming the model presented in this paper. The authors should also be more objective in their presentation of the results.

### Reproducibility

It seems like the authors did not link their code in the paper in any way, thus it's not easily possible to check whether the claimed results are reproducible.

### Presentation/Clarity

You claim that the post-processing pipeline is designed to maximize performance under limited data constraints, it is not clear to me why the specific post-processing is supposed to help with limited data specifically. I also feel like it's not clear why specific changes in the training setup, like the reduced learning rate for specific parts of the architecture are made.
In section 4.1 you state "However, our approach demonstrates competitive performance compared to the original SAM-Med3D baseline in most modalities.", this simply does not seem to be true, according to your results, the original SAM-Med3D seems to perform significantly better almost everywhere, the only case where your approach matches its performance is when it comes to the NSD metric in Microscopy where both your approach and the original SAM-Med3D have a score of 0.
Similarly the results do not seem convincing enough to claim that your post-processing pipeline is "leading to a noticeable improvement in segmentation accuracy and visual quality".

### Other comments

- You mention that Figure 5 shows 200 epochs, but the plot only shows a little more than 100, also it is a bit weird to have Figure 4 and Figure 3 both be sub-figures of Figure 5, I would recommend only having a Figure 5 there. You also mention that the smooth curves in there "indicate good generalization without overfitting behavior", unless the left plot was made on some validation set the curves do not tell you that much about whether you are overfitting or not.
- In Section 4.4 the sentence after Limited Training Data Impact seems to be cut off abruptly
- The ablation in Section 4.2 is still missing, it would have been interesting to see a comparison with the original architecture and training setup of SAM-Med3D trained on the exact same dataset as your custom version, to see the impact of your changes to the training pipeline
- In Fig 6:
	- It is hard to see any difference between the pictures with preprocessing and without, maybe another example would have been better
	- The bottom ground truth does not contain any yellow, why do the predictions contain some?
- Related Work section is quite short, could have gone into some more detail, especially for SAM-Med3D which you are building upon
- The Table reference in Section 4.1 is broken, also some explanation for why SAM-Med3D and your own variant achieve a score of 0 for NSD on Microscopy would have been good
- You mention that the training approach "includes carefully tuned hyperparameters", how were they tuned?
- Section 3.1 misses information about DSC_AUC, NSD_AUC, I would also recommend that you add information about how the 4 metrics get aggregated to compute the final ranking in the competition
- In Section 2.2 you mention that the sequence of operations was "validated for optimal performance", what do you mean by that?
- Section 2.7 is empty, if you used the 10% provided by the organizers then mention this somewhere, if you picked the 10% yourself it would be good to clarify how exactly this was done
- Section 2.8 also still contains the default text of the paper template

---

### Official Review · Reviewer_dKgW · 2025-09-15
**Review of "Enhanced SAM-Med3D: A Robust Solution for 3D Medical Image Segmentation with Advanced Post-processing"**

**Rating:** 3
**Confidence:** 4

**Review:**

### Summary:

The authors propose an extended version of SAM-Med3D for interactive segmentation of 3D medical images. Their approach introduces a five-stage post-processing pipeline and applies strategic training optimizations designed to compensate for limited training data.

### Strengths:

* Quantitative evaluation is conducted across multiple imaging modalities.

### Weaknesses:

* The official SAM-Med3D baseline achieves a DSC of 0.54 on the coreset track, which is substantially higher than the 0.28 reported for the proposed variant. This suggests that the proposed training optimizations have a markedly negative effect.
* Similarly, the five-stage post-processing pipeline appears to negatively affect results for most modalities, with the exception of PET.
* Overall, the paper gives the impression of being incomplete or not yet finalized.

### Suggestions for Improvement:

1. In Section 2.1, the related models should not only be listed but also compared, highlighting how the approaches differ from each other.
2. In Section 2.2, the subsection “Overlap Resolution” does not specify the strategy used to prioritize labels. This should be clarified.
3. In Section 2.2, the claim that *“This structured post-processing pipeline systematically enhances the raw model output, leading to a noticeable improvement in segmentation accuracy”* is misleading, as the pipeline generally worsens performance. This statement should be toned down or restricted to PET results only.
4. In Section 2.4, it is stated that the prompt encoder can also process bounding boxes. However, the baseline SAM-Med3D only supports point prompts. If the model was extended in this regard, the modifications should be described in detail.
5. The legend in Figure 2 is redundant, as the component names are already included in the boxes, and should therefore be removed.
6. Sections 2.7 and 2.8 are currently empty and should be deleted.
7. Figure 5 is missing but still referenced with a caption. It likely refers to Figures 3 and 4. Moreover, the x-axis in these figures shows only 100 epochs instead of the 200 mentioned in the description.
8. The table in Section 4.1 lacks a caption, which also leads to an error in the corresponding reference in the text.
9. The ablation study announced in Section 4.2 is missing and should be included.

---

### Official Review · Reviewer_Yq6X · 2025-09-16
**Enhanced SAM-Med3D: A Robust Solution for 3D Medical Image Segmentation with Advanced Post-processing**

**Rating:** 3
**Confidence:** 5

**Review:**

1. Summary

This paper presents an enhanced method based on SAM-Med3D, designed to tackle the challenge of 3D medical image segmentation under significant data constraints (using only 10% of training data). The authors' core contribution is the design and implementation of a five-step post-processing pipeline (region filtering, hole filling, morphological closing, Gaussian smoothing, and overlap resolution). The approach aims to refine the model's output to improve segmentation quality without modifying the SAM-Med3D model itself. The method was evaluated on a multi-modal challenge dataset, and the paper provides a detailed analysis of the post-processing effects across different modalities.

2. Strengths

1）Problem Significance: The paper addresses medical image segmentation in a few-shot learning scenario, a research direction of great practical importance and challenge. Acquiring large volumes of high-quality annotated data is often prohibitively expensive in clinical practice, making it crucial to study methods that perform well under data scarcity.

2）Systematic Approach: The authors propose a post-processing pipeline with a clear structure and well-defined steps. This modular design is easy to understand and reproduce.

3）Honest and In-depth Analysis: The most commendable aspect of this paper is its analysis of the experimental results. The authors are remarkably transparent in showing the complex effects of their pipeline across different modalities: it achieves positive results on PET images but degrades performance on CT, MRI, and others. This deep, modality-specific analysis provides valuable lessons and insights for future research.

4）Clarity of Writing: The paper is well-written overall. The descriptions of the method, experimental setup, and results are clear and easy to follow.

3. Weaknesses

1）Significant Performance Issues: This is the most critical weakness of the paper. The quantitative results (Table 12) show that the overall performance is far below the current state-of-the-art (e.g., DSC of 0.38 on CT, whereas methods like nnInteractive achieve over 0.87). More critically, the core contribution—the post-processing pipeline—actually leads to a performance drop in most cases (average DSC decreases from 0.28 to 0.27). The effectiveness of a method that makes the results worse is highly questionable.

2）Lack of Novelty: The post-processing techniques used in the paper, such as morphological operations, Gaussian smoothing, and removal of small connected components, are all standard and classic image processing operations. Combining them into a pipeline is a reasonable engineering effort, but it lacks sufficient novelty from a research perspective, especially when the combination fails to yield generalizable performance improvements.

3）Contradiction between Method and Conclusion: The authors propose a "universal" post-processing pipeline with fixed parameters, yet their experimental results and final conclusion explicitly state that a universal approach is ineffective and that an adaptive, modality-specific strategy is required. This essentially uses the results to invalidate the rationale of the proposed method. The analysis section of the paper is more valuable than the method itself.

4. Questions for the Authors

1）Given that the post-processing pipeline had a negative impact on most modalities, have you considered repositioning the core contribution of the paper? For instance, instead of presenting it as a "robust solution," perhaps frame it as a study analyzing why universal post-processing fails in a low-data segmentation regime. From that perspective, what would you say is the main takeaway from this work?

2）All parameters in the post-processing pipeline (e.g., region size threshold of 64, Gaussian sigma of 0.8) are fixed. Given the vast differences across imaging modalities and anatomical structures, was this fixed-parameter approach too simplistic? Did you attempt to perform any parameter tuning for each modality individually?

---

### Official Review · Reviewer_qYKJ · 2025-09-18
**Review of Enhanced SAM-Med3D: A Robust Solution for 3D Medical Image Segmentation with Advanced Post-processing**

**Rating:** 3
**Confidence:** 5

**Review:**

**Summary**

The authors propose an enhanced version of SAM-Med3D that incorporates five post-processing techniques after segmentation. Their work focuses specifically on the CoCa data challenge, which allowed them to use only a small portion of the training data. Their proposal does not modify the underlying neural network; instead, it adds several post-processing steps to improve the segmentation output.

**Review**

1. The use of words like "**Robust**" and "**Advanced**" in the heading is vague regarding the core idea of this work. It would be helpful if the authors could modify the title to be more specific.

2. The abstract includes a quote, "**demonstrate some improvement**," which is also very unclear. The authors need to provide quantitative values to specify the extent of the improvement.

3. As this is a competition paper, providing the code via a **GitHub link** would help reviewers better understand the claims made.

4. The phrase "**advanced post-processing**" seems unclear. How do the authors define their methods as "advanced" compared to state-of-the-art techniques? If this is not explained, it would be best for the authors to avoid using such subjective terms.

5. In Section 2.2, "Post-processing (1. Region filtering)," what is the reasoning behind choosing **64 voxels** for region filtering? The authors should also provide detailed steps on how they selected the hyperparameters for all post-processing steps.

6. Sections 2.7 and 2.8 appear to be incomplete. The authors should review their submission to ensure these sections were not inadvertently omitted.

7. In Figures 3 and 4, the authors state "200 epochs" in the text, but the figures only show up to **100 epochs**. The x-axis on the figures should be extended to include 200 epochs.

8. In Section 4.1, there is a typo: "**Table ??**." This should be corrected.

**Conclusion**

The authors propose a robust method for segmentation under a low-data paradigm, but they are unable to make strong claims based on the analysis provided. The improvement in the Dice Similarity Coefficient (**DSC**) is marginal, and there is variability with different methods. Although their method provides improved DSC values overall, it actually reduces the DSC for some modalities. Consequently, the authors' claim that their method is **robust** is not fully supported by the details provided in the paper.

---

### Official Review · Reviewer_qM3X · 2025-09-28
**Incremental Post-Processing Add-On without Convincing Contribution**

**Rating:** 3
**Confidence:** 4

**Review:**

This paper presents an “Enhanced SAM-Med3D” approach for the CVPR 2025 3D Medical Image Segmentation Challenge, specifically under the coreset track with only 10% training data. The authors adopt SAM-Med3D as the base model and mainly propose a five-step post-processing pipeline (region filtering, hole filling, morphological operations, Gaussian smoothing, and overlap resolution). Experiments are conducted across multiple imaging modalities (CT, MRI, PET, Ultrasound, Microscopy). The results show limited improvements, with slight gains in PET but performance degradation in most other modalities. The claimed contribution is demonstrating how targeted post-processing can help under data-constrained conditions.

Major Issues

1. There are frequent redundancies and overlong descriptions (e.g., pages are spent re-explaining SAM-Med3D without clear new insight). Moreover, basic academic writing problems exist: missing clarity in motivation, inconsistent use of notation (e.g., tables with “??” placeholders), awkward phrasing, and incomplete subsections (e.g., Section 2.7/2.8 left as “if available”).
2. The motivation is unclear. Why focus primarily on post-processing rather than core model modifications? The paper spends disproportionate effort describing SAM-Med3D itself, which is prior work. Claimed novelty (the 5-step post-processing) is incremental and heuristic. These steps (region filtering, hole filling, Gaussian smoothing, etc.) are well-known in medical image processing and do not constitute a significant methodological advance.
3. Reported results are weak: the proposed method achieves average Dice 0.28 (w/o PP) and 0.27 (w/ PP), which is substantially below competing methods (e.g., VISTA3D, nnInteractive). The ablation study is missing. Section 4.2 explicitly says results “will be updated once validation is complete,” which is unacceptable for a workshop submission. No qualitative analysis beyond a single figure, and even their improvements are not convincing. The negative results of post-processing on most modalities are not adequately addressed, this undermines the main claim.
4.  The pipeline is fixed and not adaptive to different modalities, yet results show modality-specific failures. The authors acknowledge this but do not provide a solution. Heavy reliance on heuristic morphological operations without theoretical or data-driven justification.
Overall, the paper suffers from serious writing quality issues, unclear motivation, missing ablation experiments, and weak/negative results. The technical contribution is minimal, relying on standard post-processing heuristics with little novelty. Given these shortcomings, I recommend a clear rejection.

---

### Decision · Program_Chairs · 2025-11-12

Revision